

# The fungal community in non-rhizosphere soil of *Panax ginseng* are driven by different cultivation modes and increased cultivation periods

Yu Bao[1,*], Bao Qi[2,*], Wei Huang[3], Bao Liu[1] and Yu Li[2]

[1] Key Laboratory of Molecular Epigenetics of the Ministry of Education, Northeast Normal University, Changchun, People's Republic of China

[2] Engineering Research Center of Chinese Ministry of Education for Edible and Medicinal Fungi, Jilin Agricultural University, Changchun, People's Republic of China

[3] Key Laboratory of Applied Statistics of Ministry of Education, Northeast Normal University, Changchun, People's Republic of China

[*] These authors contributed equally to this work.

Corresponding authors
Bao Liu, baoliu@nenu.edu.cn
Yu Li, fungi966@126.com

## ABSTRACT

Continuous cropping obstacles severely hindered the sustained development of the ginseng industry. Among the obstacles, an imbalance of soil microbiome community was considered one of the major culprits. The fungal community is an essential part of the soil microbiome community. Extensive characterization of the fungal community composition and variation during ginseng cultivation will help us understand the mechanism underlying continuous cropping obstacles. By using a high-throughput amplicon sequencing method, the non-rhizospheric fungal community of farmland cultivated ginseng of 2 years old (C2) and 5 years old (C5), understory wild ginseng of 15 years old (W15) and 35 years old (W35), fallow fields which have been abandoned for 10 (F10) years were characterized. Farmland cultivated ginseng and understory wild ginseng harbored distinct non-rhizospheric fungal communities, and extension of cultivation periods enlarged the fungal community difference between two cultivation modes. Extended cultivation periods significantly decreased the OTU richness and PD whole tree indices, and OTU number and cultivation periods were negatively correlated. Extension of cultivation periods led to an increased abundance of pathotrophs. Still, the increased abundance of pathotrophs may not be the leading cause of severe continuous cropping obstacles in farmland cultivated ginseng. Compared with understory wild ginseng, farmland cultivated ginseng had a lower abundance of symbiotrophs and a higher abundance of saprotrophs. This changed symbiotrophs/saprotrophs ratio may have some correlation with the severe continuous cropping obstacles that occurred in farmland cultivated ginseng. Fallowing on the fungal community of the non-rhizosphere soil was generally opposite of that of extension of ginseng cultivation periods. The impacts of farmland cultivation on the fungal community of the non-rhizosphere soil can last for decades, even if the following is practiced.

## INTRODUCTION

Ginseng (*Panax ginseng* C.A. Mey.) has been used as a traditional Chinese herbal medicine for thousands of years. Nowadays, the global market value of ginseng is about 3.5 billion US dollars annually, and the demand for ginseng products is still rising. Cultivated ginseng provides the majority of the market demand, which primarily includes two cultivation modes, farmland cultivation (cultivated ginseng) and understory cultivation (wild ginseng). One of the most severe hindrances of ginseng cultivation is continuous cropping obstacles. Ginseng cannot be consecutively cultivated in the same field without causing severe loss of yield and quality. It has been reported that the replantation of ginseng may cause up to 75% of ginseng seedling death (*Wu et al., 2008*). Usually, more than 30 periods of crop rotation were needed for successful replanting (*Yang et al., 2004*), which makes proper soils suitable for ginseng cultivation a scarce resource. Farmland cultivation of ginseng has led to large-scale deforestation in the northeast region of China, which is unsustainable and has severely hindered the sustained development of the ginseng industry.

Several factors have been shown or implicated to correlate with continuous cropping obstacles, one of which is an imbalance of soil microbiome community (*Ogweno & Yu, 2006*; *Wu et al., 2008*). Being an essential part of the soil microbiome community, soil fungal community changes during ginseng cultivation had received much attention, and it has been shown that rhizospheric fungi communities can be affected by different cultivation ages, growth stages and cultivation modes (*Li et al., 2012*; *Liu, 2013*; *Xiao et al., 2016*; *Dong et al., 2018*). However, these studies mainly focused on the rhizosphere compartment of soil, which refers to the narrow zone surrounding the root (*Estabrook & Yoder, 1998*). For most of the soil, namely the non-rhizosphere compartment of soil, little work has been done (*Li et al., 2012*; *Liu, 2013*). Microbial community in ginseng non-rhizosphere and rhizosphere soil were reported to be different, and with the increase of cultivation ages, this difference became more significant (*Li et al., 2012*; *Liu, 2013*). However, it is not clear whether the microbial communities in non-rhizosphere soil can also be affected by ginseng cultivation. Meanwhile, due to the technical short come of Random Amplified Polymorphic DNA (RAPD) and denaturing gradient gel electrophoresis (DGGE), these two studies did not provide many details of the fungi communities dynamics in non-rhizosphere soil (*Li et al., 2012*; *Liu, 2013*). More extensive characterization of the non-rhizospheric fungal community composition and variation during ginseng cultivation is essential to understand the underlying mechanism of continuous cropping obstacles.

In contrast to farmland cultivated ginseng, understory wild ginseng was grown in the natural forest. It can be consecutively cultivated in the same forest for a long time without causing severe continuous cropping obstacles. Illustrating the fungal community difference between understory wild ginseng and farmland cultivated ginseng will provide insights into the mechanism underlying continuous cropping obstacles. Land fallow is a practical approach to achieve replantation, but the 30–40 periods of rotation time needed is too long. Extensive characterization of the non-rhizospheric fungal community dynamics during land fallow will help us understand the mechanisms underlying crop rotation and provide clues to accelerate this process.

The present study was designed to characterize the influence of different cultivation ages and cultivation modes to the non-rhizospheric fungal community of ginseng, with the hope to provide insights into the mechanism underlying continuous cropping obstacles.

## MATERIALS & METHODS

### Soil sample collection and DNA extraction

Soil samples were collected in growth season (July 10 th 2016) from Dadong Village (42.31 N 127.19 E), Fusong, Jilin Province of China. Fusong is the central ginseng production region of China, and the Dadong Village is one of the several places where understory wild ginseng has been grown for over 40 years. Soil samples were collected from fields of farmland cultivated ginseng of 2 years old (C2) and 5 years old (C5), understory wild ginseng of 15 years old (W15), and 35 years old (W35), fallow fields which have been abandoned for 10 (F10) years after five years of ginseng farmland cultivation. Permission to access the field site was obtained from the landowner, Shihua Chen. We chose 2-year-old and 5-year-old farmland cultivated ginseng because after two years of growth, disease occurrence and death rates of farmland cultivated ginseng generally increase (*Dong et al., 2018*) and farmland cultivated ginseng is usually harvested after five years of farmland cultivation. Thirty-five-year-old understory wild ginseng was the oldest understory wild ginseng we can found. Coupled with the 15-year-old understory wild ginseng, we can investigate the long-term influence of understory wild ginseng cultivation on the non-rhizospheric fungal community.

The understory wild ginseng used in this study was direct planted to the natural forest and grown without any manual intervention. The soil used for farmland cultivation was removed from the same area of natural forest where understory wild ginseng was grown. Fallow fields used in this study have been abandoned for ten years without planting any crops to avoid the influence of crop cultivation practice on soil microbial community.

Non-rhizosphere soil components of ginseng (20 cm away from ginseng plants) were collected as study materials with three biological replications. For each sample, 40 soil cores were collected from about 2,500 m$^2$ circular plots and pooled together. Each soil core was collected using a five cm diam PVC tube, which was hammered into the soil and collect the soil from the surface to down to five cm depth (*Tedersoo et al., 2014*). Power Soil DNA Isolation kit (MoBio, Carlsbad, CA USA) was used to extracted DNA from about 10g soil according to the manufacturer instructions.

### Illumina sequencing of ITS rRNA gene amplicons

Fungi intergenic transcribed sequence (ITS) universal primers F2045 (5′–GCATCGATGAAGAACGCAGC-3′) and R2390 (5′- TCCTCCGCTTATTGATATGC-3) were used. Library construction and sequencing were carried out by Realbio Genomics Institute (Shanghai, China). Illumina Hiseq 2500 platform was used to perform amplicon sequencing and generated about 100100,000 250 bp paired-end (PE) reads in each sample. Original data generated in this study have been deposited into the NCBI SRA database, and the accession number was PRJNA523683.

## Data analyses

PANDAseq (*Masella et al., 2012*) was used to assemble the paired-end (PE) reads, quality assessment and filtering were carried out with QIIME1 pipeline with default parameters (*Caporaso et al., 2010*). Chimeric sequences were identified and filtered by the UCHIME program (*Edgar et al., 2011*). USEARCH implemented in QIIME1 software (*Sachs, Skophammer & Regus, 2011*) was utilized to assign Operational taxonomic units (OTU) at the 97% sequence identity level. The most abundant sequence of each OUT was chosen as the representative sequence and then taxonomically classified using the RDP classifier against the Ribosomal Database Project (RDP) database (*Wang et al., 2007*; *Brown et al., 2013*). OTUs which belong to mitochondrion or chloroplast and singleton OTUs were removed. Alpha and beta diversities were calculated using the QIIME1 pipeline (*Caporaso et al., 2010*) after rarefied to 132,764 sequences for each sample. We used linear discriminant analysis effect size (LEfSe) (*Segata et al., 2011*) to identify the differentiation of fungal communities between different samples at multiple taxonomical levels, Kruskal-Wallis test $P < 0.05$ and LDA score of >2 were used as thresholds. Principal coordinate analyses (PCoA) based on unweighted UniFrac distance metrics were carried out using the QIIME1 pipeline (*Caporaso et al., 2010*). ADONIS was carried out using the R package vegan (*Dixon, 2003*).

Differences of alpha diversity indexes between samples were determined by Tukey's method (multiple comparisons of means), and normality and homogeneity of variance analysis were respectively performed by using the Shapiro–Wilk normality test and Brarlett test. These analyses were carried out using R software (Version 3.5.2, https://www.r-project.org/). OTUs were parsed against the FUNGuild database to assign putative trophic strategies (*Nguyen et al., 2016*).

## Regression analysis

We used a linear model (LM) regression analysis to test the relationship between OTU richness and cultivation periods. Standardized major axis (SMA) analysis was carried out using (S)MATR package in R (*Falster, Warton & Wright, 2006*; *Warton et al., 2011*) to fit a line of best fit and to obtain a slope estimate and R2 value.

# RESULTS

## Variation in fungi community compositions

Soil samples included four cultivation periods: 2-year-old (C2) and 5-year-old (C5) cultivated ginseng, 15-year-old (W15), and 35-year-old (W35) understory wild ginseng. Intergenic transcribed sequence (ITS) gene amplicon sequencing generated 135,340-189,257 reads and was rarefied to 132,764 clean reads for each sample. The Good's coverage scores ranged from 0.996792 to 0.999862, indicating that the sequencing depth was sufficient to quantify the fungi communities (Table 1).

According to Principal Co-ordinates Analysis (PCoAs) analyses, samples from farmland cultivated ginseng and understory wild ginseng showed clear separation (Fig. 1A). PCoAs1 and PCoAs2 explained 23.13% and 16.68% of variation, respectively (Fig. 1A). ADONIS analysis showed that fungi communities in non-rhizosphere soil of farmland cultivated

**Table 1  Statistic of alpha diversity indexes in each sample.**

| Alpha name | OTUs | PD whole tree | Simpson | Goods coverage |
|---|---|---|---|---|
| W15 | 1,341 ± 265.99 | 331.4288 ± 46.1156 | 0.9688 ± 0.0054 | 0.9993 ± 0.0004 |
| W35 | 421 ± 169.01 | 137.432 ± 41.1016 | 0.9759 ± 0.0097 | 0.9998 ± 0.0001 |
| C2 | 2,106.7 ± 156.7 | 408.7553 ± 33.5402 | 0.9624 ± 0.0113 | 0.9973 ± 0.0004 |
| C5 | 1,031 ± 98.02 | 239.2621 ± 17.9476 | 0.9523 ± 0.0182 | 0.9980 ± 0.0002 |
| F10 | 1,812.3 ± 200.9 | 427.1849 ± 34.8345 | 0.9827 ± 0.0054 | 0.9976 ± 0.0003 |

**Notes.**
C2, farmland cultivated ginseng of 2 years old; C5, farmland cultivated ginseng of 5 years old; F10, fallow fields which have been abandoned for 10 years after 5 years of farmland cultivation; W15, understory wild ginseng of 15 years old; W35, understory wild ginseng of 35 years old.

ginseng and understory wild ginseng were significantly different ($R^2 = 0.22377$, $P = 0.003$). Only 27.4% of OTUs from farmland cultivated ginseng and understory wild ginseng were overlapped (1,494 out of 5,446) (Fig. 1B). There was 1,310 common OTUs (28.1% of a total of 4,663 OTUs in C2 and W15) between C2 and W15, the number of common OTUs between C5 and W35 decreased to 285 (11.8% of a total of 2414 OTUs in C5 and W35, Fig. 1B). These results indicated that with the extension of cultivation periods, the differences of fungal community between the two cultivation modes were enlarged. In both farmland cultivated ginseng and understory wild ginseng, Ascomycota and Basidiomycota were dominant phyla, which accounted for 75.1%-91.1% of the total OTUs (File S1). LEfSe analysis showed that Agaricomycetes, Pezizomycetes and Glomeromycota were significantly enriched in understory wild ginseng compared to that of cultivated ginseng. In contrast, Sordariomycetes, Dothideomycetes, Ascomycota_unidentified, Eurotiomycetes, Tremellomycetes, Lecanoromycetes, Cystobasidiomycetes, and Wallemiomycetes were significantly enriched in farmland cultivated ginseng compared to that of understory wild ginseng (Figs. 1C, 1D, LEfSe, $P < 0.05$, LDA>2).

## Extension of cultivation periods lead to decreased OTU richness and PD whole tree indices in both cultivation modes

Extension of cultivation periods significantly decreased the OTU richness and PD whole tree indices in both understory wild ginseng and farmland cultivated ginseng (Tukey's method, $P < 0.01$; Table 1). After another three years of farmland cultivation, the OTU number decreased from 3261 OTUs in C2 to 1724 in C5, which was 52.9% of that in C2 (5914 OTUs) (Fig. 1B, File S1). Similar results were also found in understory wild ginseng, coupled with the extension of cultivation periods, the OTU number decreased from 2712 in W15 to 975 in W35 (Fig. 1B, File S1). The OTU number and cultivation periods were negatively correlated (SMA, $p$-value = 0.0020962, R2 = 0.62863, slope = -48.42971, 95% CI's = $-73.50041$ to $-31.91053$, Fig. 2). However, the species evenness was not affected by the extension of cultivation periods in both cultivation modes, as indicated by Simpson indices (Tukey's method, $P > 0.05$, Table 1).

With the extension of cultivation periods, non-rhizosphere soil of more extended cultivation periods lost relatively the same amount of OTUs in the two cultivation modes (2000 OTU loss in W35 and 2241 loss in C5), while non-rhizosphere soil of farmland
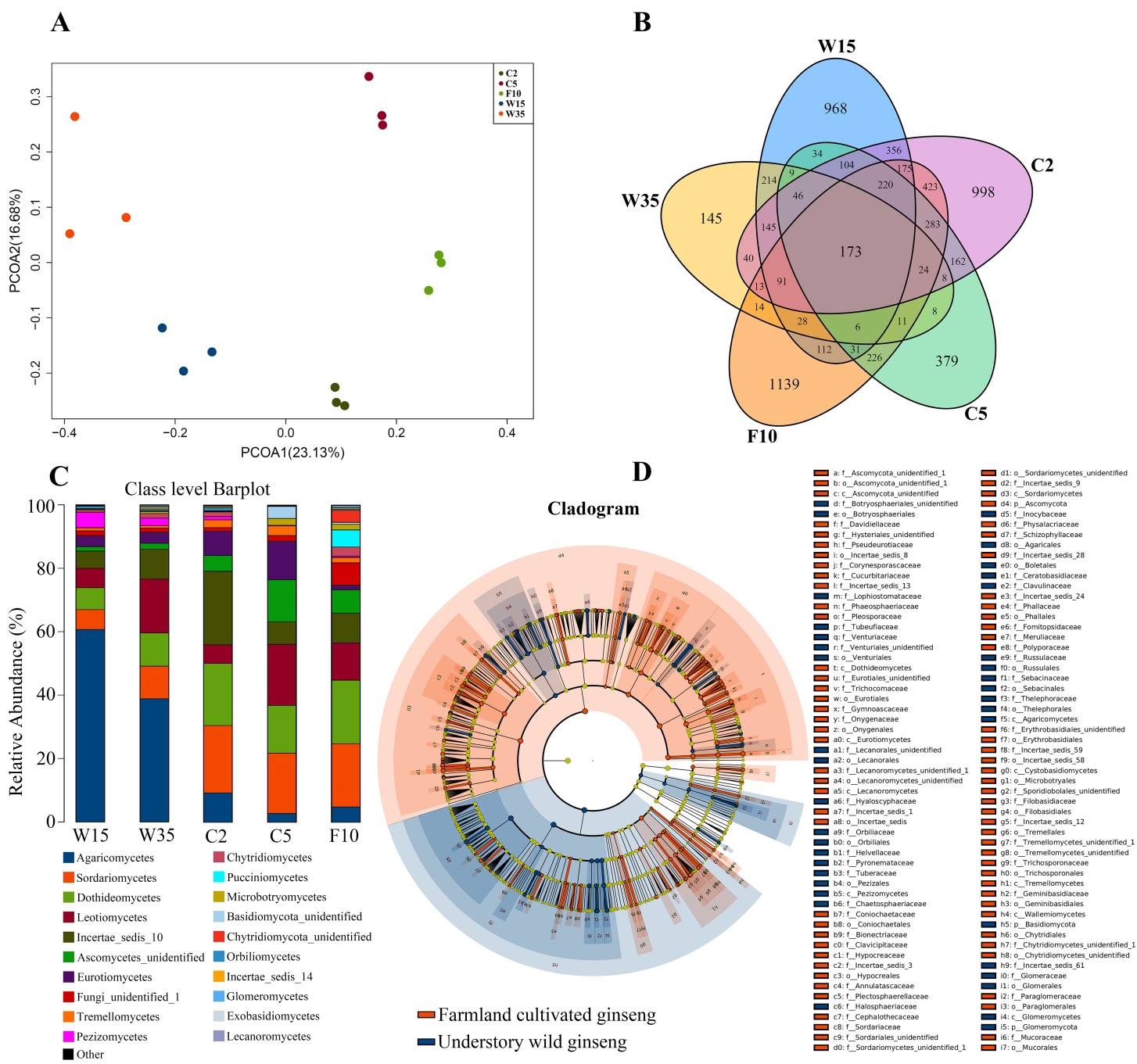

**Figure 1** **Non-rhizosphere soil associated fungal community.** (A) Distances show separation of Principal coordinate analysis (PCoA) plot using Unweighted UniFrac method. (B) Venn diagrams illustrating the number of those OTUs that are common between farmland cultivated ginseng and understory wild ginseng samples. (C) Taxonomic classification and relative abundance of fungal community of each sample at the class level. (D) LEfSe comparison of fungi community between non-rhizosphere soil of farmland cultivated ginseng and understory wild ginseng. Cladogram represented the hierarchical structure of differently abundant taxa. The taxonomic levels from phylum to family are labelled.

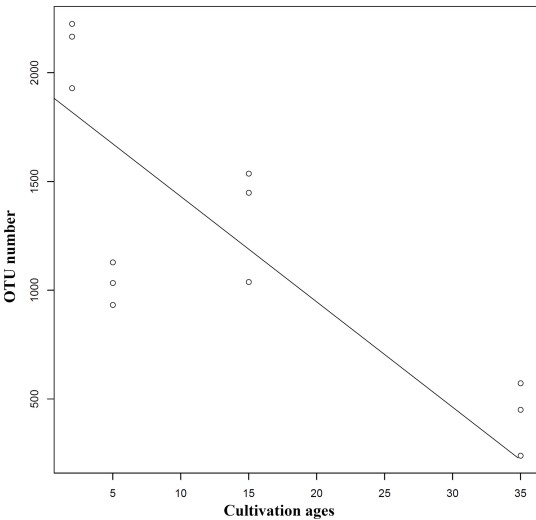

**Figure 2** **Regression analyses showing the negative relationship between OTU number and cultivation period.** The correlation was statistically significant (SMA, *p*-value = 0.0020962, R2 = 0.62863, slope = −48.42971, 95% CI's [−73.50041−−31.91053]). Lines of best fit was indicated using SMA models.

cultivated ginseng acquired more OTUs than that in understory wild ginseng (704 OUT gain in C5 compared with 263 in W35, Fig. 1B). Only a small proportion of the lost or gained OTUs in more extended cultivation periods of the two cultivation modes were overlapped (Fig. 1B). Specifically, there was 531 common OUT loss (26.6% in W35 and 23.7% in C5) in samples of more extended cultivation periods, and the number of common OUT gain in samples of more extended cultivation period was 19 (7.2% in W35 and 2.7% in C5, Fig. 1B).

Under farmland cultivation mode, with the extension of cultivation periods the abundance of Leotiomycetes, Ascomycota_unidentified, Microbotryomycetes, and Agaricostilbomycetes were significantly increased, while the abundance of Agaricomycetes, Incertae_sedis_10, Pezizomycetes, Chytridiomycetes, Orbiliomycetes, Incertae_sedis_14, Glomeromycetes, and Saccharomycetes were significantly decreased (Fig. 3, LEfSe, *P* < 0.05, LDA>2). Under understory wild ginseng cultivation mode, with the extension of cultivation periods, the abundance of Incertae_sedis_14 was significantly increased, and the abundance of Chytridiomycetes_unidentified was significantly decreased in non-rhizosphere soil compartment (Fig. 4, LefSe, *P* < 0.05, LDA>2).

### Influence of fallowing on non-rhizospheric fungal community

Ginseng plants were harvested after five years of farmland cultivation, and the resulted fields were abandoned for ten years without growing any crops. According to PCoAs analyses, samples from fallow soil and farmland cultivated ginseng showed clear separation (Fig. 1A). ADONIS analysis showed that non-rhizospheric fungi communities in the fallow field and farmland cultivated ginseng were significantly different ($R^2 = 0.35168$, $P = 0.009$). After fallowing for ten years, the OTU richness, PD whole tree indices, and species evenness in non-rhizosphere soils were significantly increased (Tukey's method, $P < 0.05$; Table 1).
**Cladogram**

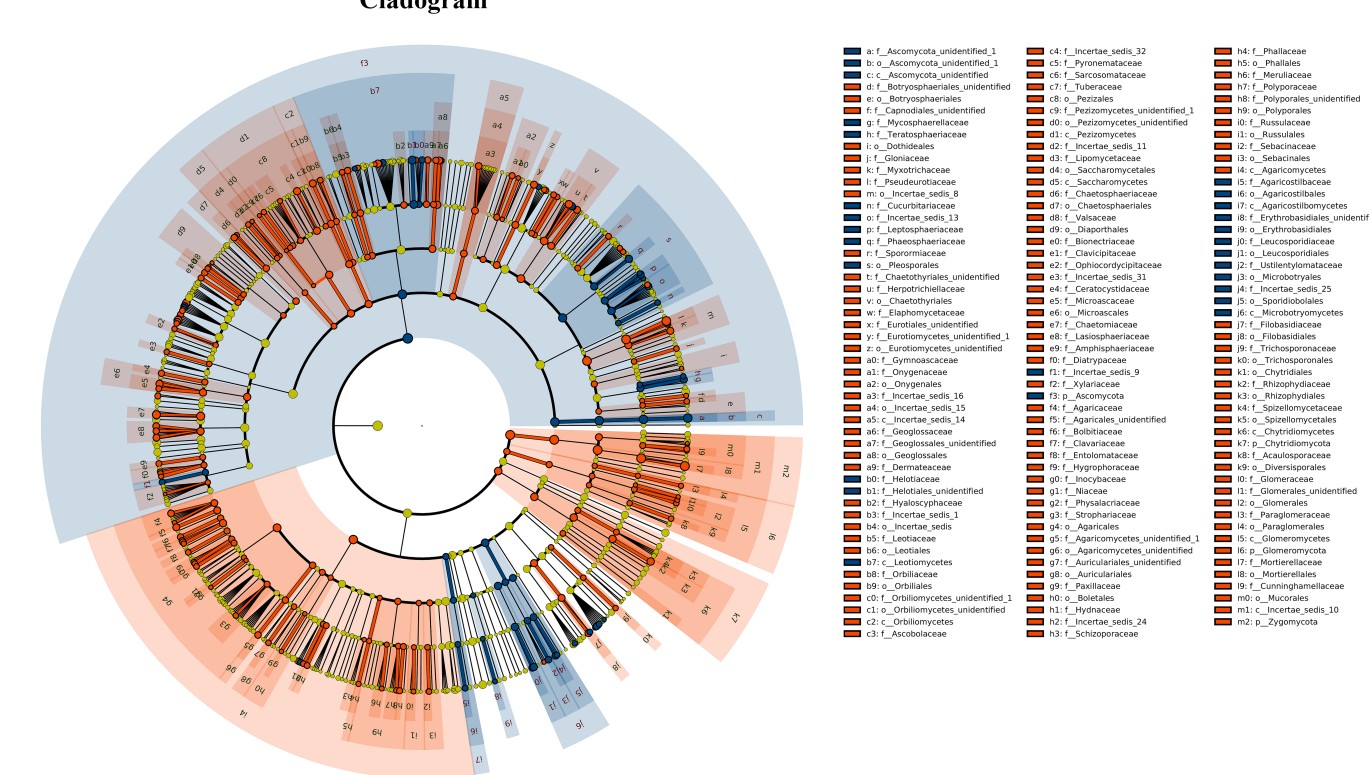

**Figure 3 LEfSe comparison of fungi community between C2 and C5.** The cladogram represented the hierarchical structure of differently abundant taxa. The taxonomic levels from phylum to family are labeled. C2, farmland cultivated ginseng of 2 years old; C5, farmland cultivated ginseng of of 5 years old.

The OTU richness in F10 recovered to 2969 OTUs, but still less than that of C2 (Fig. 5A, File S1). Fallowing recovered 31.3% (702 out of 2,241) of the lost OTUs in C5 during the extension of cultivation periods, and for the gained OTUs, 38.2% (274 out of 704) of which were conserved in F10 (Fig. 5A). The other 1293 OTUs were gained OTUs in F10 (Fig. 5A). These results demonstrated that an increase of OTU richness in fallowed soil was achieved mainly by the gain of new OTUs other than by recovering the lost OTUs during the extension of cultivation periods. Compared with C5, the abundance of Pezizomycetes, Chytridiomycetes, Pucciniomycetes, Chytridiomycetes_unidentified, Orbiliomycetes, Incertae_sedis_14, and Glomeromycetes were significantly increased, and the abundance of Eurotiomycetes, Ascomycota_unidentified, Lecanoromycetes, Cystobasidiomycetes, and Agaricostilbomycetes were significantly decreased in F10 (Fig. 5B, LefSe, $P < 0.05$, LDA>2).

## Trophic mode prediction

FUNGuild software was used to assign a putative life strategy. Compared with understory wild ginseng, the non-rhizosphere soil of farmland cultivated ginseng had a lower abundance of symbiotrophs (Fig. 6). What is more, with the extension of the cultivation period, the abundance of symbiotrophs became even lower (Fig. 6). After fallowing for ten

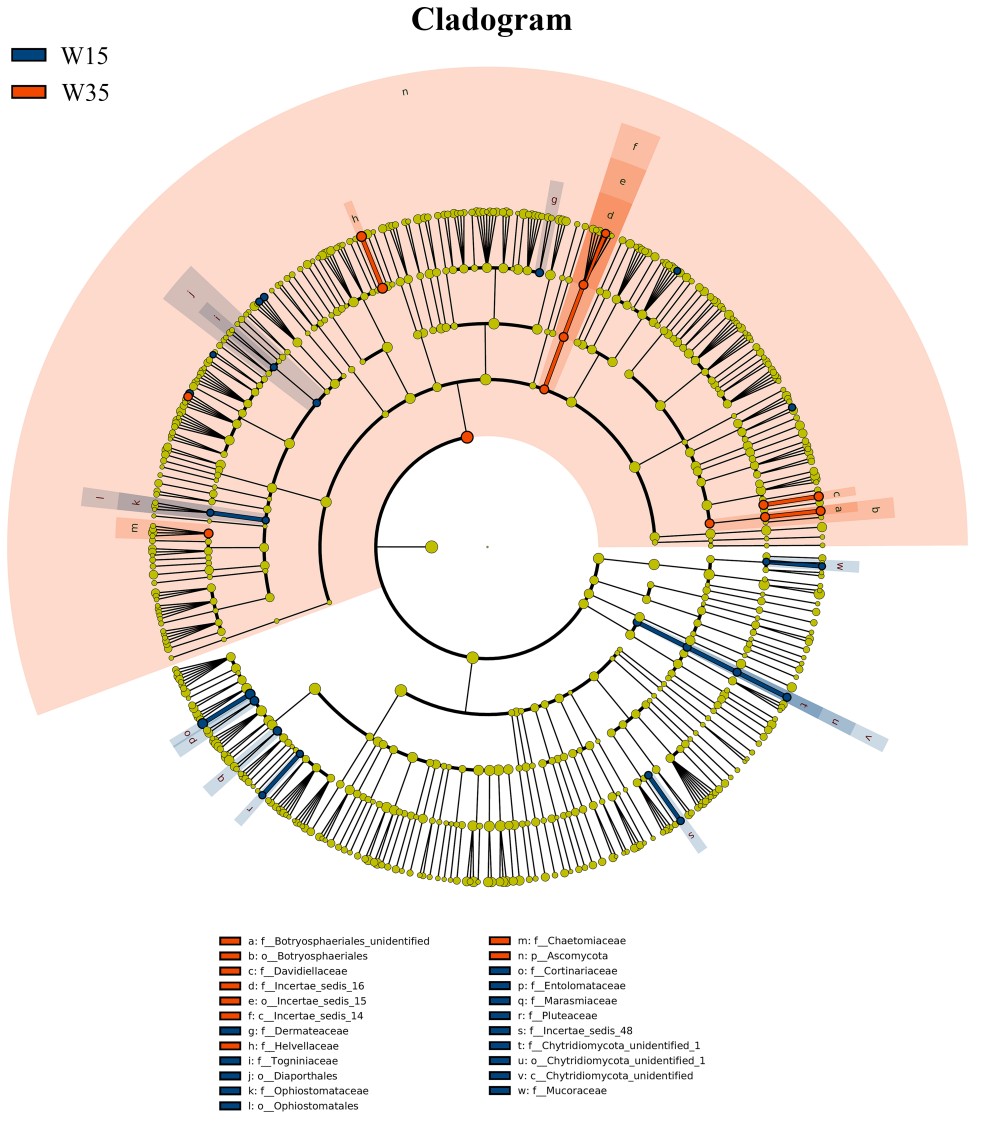

**Figure 4** **LEfSe comparison of fungi community between W15 and W35.** The cladogram represented the hierarchical structure of differently abundant taxa. The taxonomic levels from phylum to family are labeled. W15, understory wild ginseng of 15 years old; W35, understory wild ginseng of 35 years old.

years, the abundance of symbiotrophs in F10 recovered and became slightly higher than C2 (31.02% in F10 vs. 29.37% in C2). In contrast, the non-rhizosphere soil of farmland cultivated ginseng had a higher abundance of saprotrophs than that of understory wild ginseng (Fig. 6). Moreover, with the extension of the cultivation period, the abundance of saprotrophs became even higher (Fig. 6). Similar to the result of symbiotrophs, the abundance of saprotrophs in F10 became even lower than C2 (42.27% in F10 vs. 56.49% in C2). Extension of cultivation period leads to the increased abundance of pathotrophs in both understory wild ginseng and farmland cultivated ginseng. F10 has the highest abundance of pathotrophs.
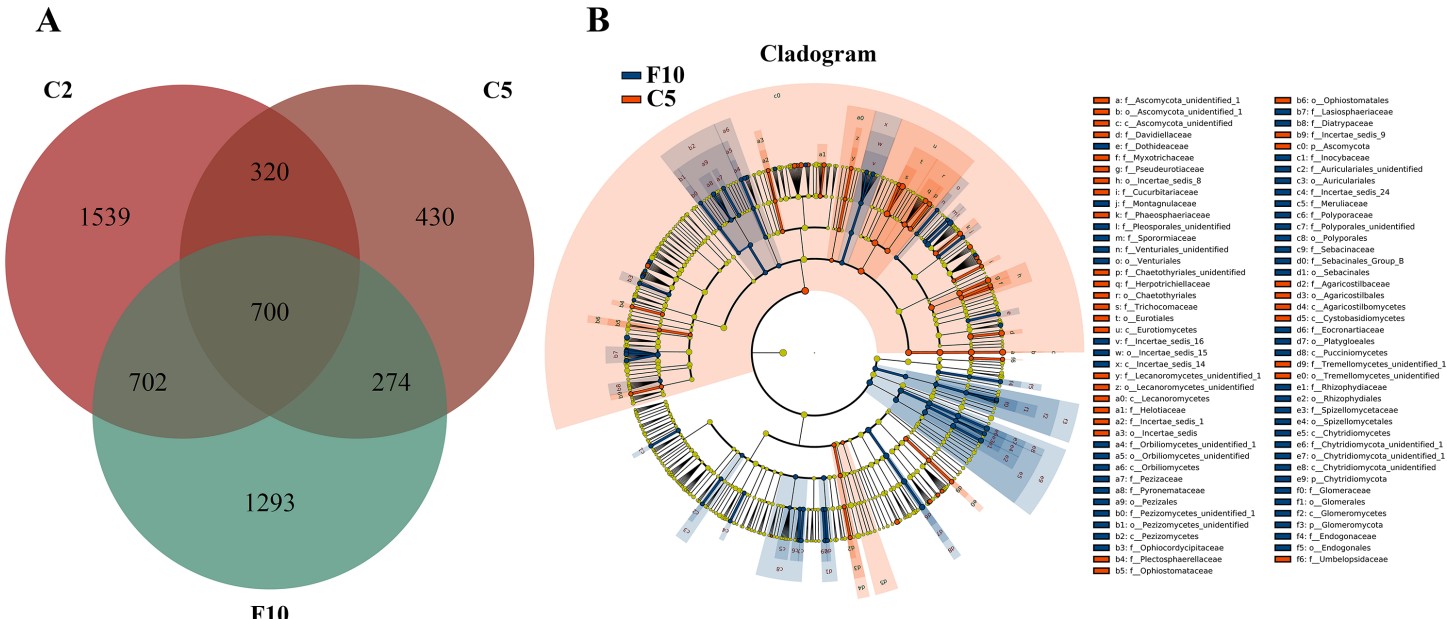

**Figure 5** **Non-rhizosphere soil fungal community variation between farmland cultivated ginseng and fallow field.** (A) Venn diagrams illustrating the number of those OTUs that are common between samples of farmland cultivated ginseng soil and fallowed soil. (B) LEfSe comparison of fungi community between F10 and C5. The cladogram represented the hierarchical structure of differently abundant taxa. The taxonomic levels from phylum to family are labeled. C2, farmland cultivated ginseng of 2 years old; C5, farmland cultivated ginseng of 5 years old; F10, fallow fields which have been abandoned for ten years after five years of farmland cultivation; W15, understory wild ginseng of 15 years old; W35, understory wild ginseng of 35 years old.

## DISCUSSION

Although the soil used to grow farmland cultivated ginseng was removed from the same forest where understory wild ginseng was grown, fungi communities in non-rhizosphere soil of farmland cultivated ginseng were significantly different from that of understory wild ginseng (ADONIS analysis, $R^2 = 0.22377$, $P = 0.003$). Only 27.4% (1494 out of 5446) of OTUs between farmland cultivated ginseng and understory wild ginseng were overlapped (Fig. 1B). In previous studies, it had been shown that rhizospheric fungi communities are affected by different ginseng cultivation modes (*Li et al., 2012*; *Liu, 2013*; *Xiao et al., 2016*; *Dong et al., 2018*). The data presented here proved that fungi communities of non-rhizosphere soil could also be affected by different cultivation modes and different cultivation ages, which means that ginseng can affect a significant part of nearby soil other than just influence the narrow zone tightly surrounding the root namely rhizosphere.

In both understory wild ginseng and farmland cultivated ginseng, extended cultivation periods significantly decreased the OTU richness and PD whole tree indexes of non-rhizosphere soil, and the OTU number and cultivation period were negatively correlated (SMA $P <0.05$). Although being different compartments of soil and containing significantly different fungal diversity index (*Liu, 2013*), similar results have been reported in rhizosphere soil of farmland cultivated ginseng in the vegetative stage (*Dong et al., 2018*). In *Panax notoginseng*, a similar result in rhizosphere soil has also been reported (*Dong et al., 2017*).

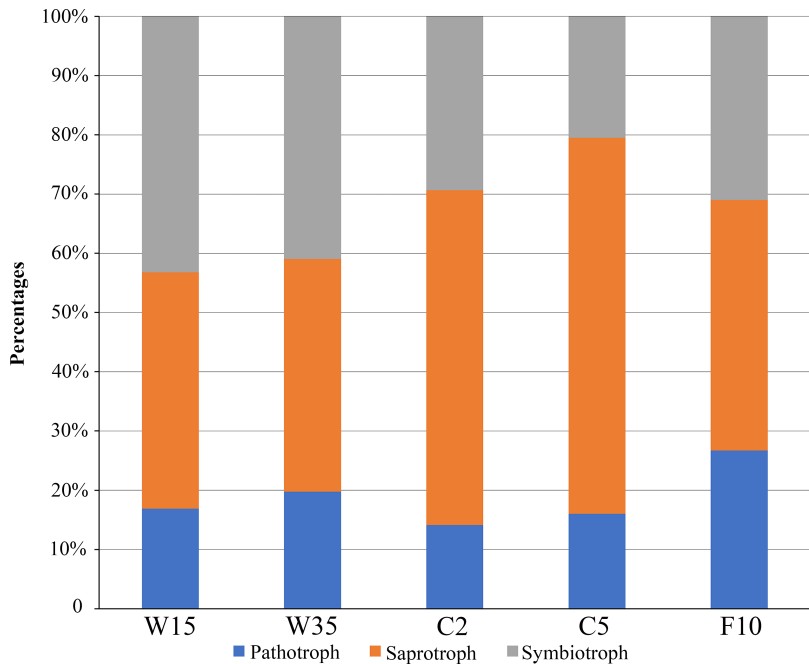

**Figure 6** **Relative abundance of fungal functional groups.** Putative functional groups were inferred by parsing against the FUNGuild database. C2, farmland cultivated ginseng of 2 years old; C5, farmland cultivated ginseng of 5 years old; F10, fallow fields which have been abandoned for ten years after five years of farmland cultivation; W15, understory wild ginseng of 15 years old; W35, understory wild ginseng of 35 years old.

Farmland cultivated ginseng cannot be consecutively cultivated in the same field without causing severe loss of yield and quality. In contrast, understory wild ginseng can be consecutively cultivated in the same forest. The 35-year-old (W35) understory wild ginseng had the lowest OUT number and PD whole tree index of all samples. This fact implied that fungal community composition changes rather than decreased fungal diversity might underly the different responses of the two cultivation modes when facing continuous cropping. Despite having the lowest alpha diversity, the non-rhizosphere fungal community of 35-year-old (W35) understory wild ginseng may still maintain some kind of balance, while in farmland cultivated ginseng, this balance may have been interrupted.

Coupled with the decreased fungi diversity, extension of cultivation periods led to an increased abundance of pathotrophs in both understory wild ginseng and farmland cultivated ginseng (Fig. 3). This result resembled the previous reports that there is a positive correlation between fungal diversity and the suppression of soil-borne plant pathogens (*Nitta, 1991*; *Garbeva et al., 2006*). However, the increased abundance of pathotrophs in farmland cultivated ginseng may not be the leading cause of severe continuous cropping obstacles in farmland cultivation ginseng, because understory wild ginseng had an even higher abundance of pathotrophs than farmland cultivated ginseng.

Compared with understory wild ginseng, farmland cultivated ginseng had a lower abundance of symbiotrophs, and with the extension of cultivation periods, the abundance

of symbiotrophs became even lower (Fig. 3). Meanwhile, compared with understory wild ginseng, farmland cultivated ginseng had a higher abundance of saprotrophs, and with the extension of cultivation periods, the abundance of saprotrophs became even higher (Fig. 3). This changed symbiotrophs/saprotrophs ratio may have some correlation with the severe continuous cropping obstacles that occurred in farmland cultivated ginseng compared with understory wild ginseng.

The effects of fallowing on the fungal community of the non-rhizosphere soil were generally opposite to that of extension of ginseng cultivation periods. Fallowing significantly changed the fungal community composition but in the opposite direction compared to that of extension of ginseng cultivation periods. Extension of cultivation periods significantly increased the abundance of Agaricostilbomycetes and Ascomycota_unidentified, while significantly decreased the abundance of Pezizomycetes, Chytridiomycetes, Orbiliomycetes, Incertae_sedis_14, and Glomeromycetes. In contrast, compared with C5, the abundance of Agaricostilbomycetes and Ascomycota_unidentified was significantly decreased, and the abundance of Pezizomycetes, Chytridiomycetes, Orbiliomycetes, Incertae_sedis_14, and Glomeromycetes were significantly increased in fallowed soil (F10). Fallowing significantly increased the fungal OTU richness and PD whole tree indices, which were also opposed to that of extension cultivation periods. Similar results have been reported in studies carried out on bacteria microbiome (Liu et al., 1983; Li, Jin & Jia, 2011).

It should be noted that the overall OTU number in F10 was still less than that in C2, and the increase of OTU richness was achieved mainly by the gain of new OTUs (1293 OTUs) other than recovering the lost OTUs (702 OTUs) during the extension of cultivation period. After ten years of soil fallowing, 38.9% (274 out of 704) newly emerged OTUs in C5 relative to C2 were still existed in F10, and only 31.3% (702 out of 2241) of the lost OTUs in C5 relative to C2 reappeared in F10. These results indicate that the effects of ginseng farmland cultivation on the fungal community of the non-rhizosphere soil can last for decades, and ten years of fallow only partly restored the fungal community changes caused by farmland cultivation. This result is consistent with the farmer's experience that usually 30–40 years of crop rotation was needed to achieve successful replantation (Yang et al., 2004).

## CONCLUSIONS

Non-rhizosphere soil of farmland cultivated ginseng and understory wild ginseng harbored distinct fungal communities, and with the increase of cultivation periods, these differences were enlarged. In both understory wild ginseng and farmland cultivated ginseng, the extension of cultivation periods significantly decreased the OTU richness and PD whole tree indices. Extension of cultivation periods led to an increased abundance of pathotrophs, but the increased abundance of pathotrophs may not be the leading cause of severe continuous cropping obstacles in farmland cultivated ginseng. Compared with understory wild ginseng, farmland cultivated ginseng had a lower abundance of symbiotrophs and a higher abundance of saprotrophs. This changed symbiotrophs/saprotrophs ratio may have some correlation with the severe continuous cropping obstacles that occurred in farmland cultivated ginseng. The effect of fallowing on the fungal community of the non-rhizosphere

soil was generally opposite to that of extension of ginseng cultivation periods. The effects of farmland cultivation on the fungal community of the non-rhizosphere soil can last for decades, even if the following is practiced.

## ACKNOWLEDGEMENTS

The authors would like to thank Shihua Chen for her assistance in the soil sampling.

### Funding

This work was supported by the University S&T Innovation Platform of Jilin Province for Economic Fungi (no. 2014B-1) and the Development Plan Project of Jilin Provincial Science and Technology Department 20160520105JH. The funders had no role in study design, data collection and analysis, decision to publish, or preparation of the manuscript.

### Grant Disclosures

The following grant information was disclosed by the authors:
University S&T Innovation Platform of Jilin Province for Economic Fungi: 2014B-1.
Jilin Provincial Science and Technology Department: 20160520105JH.

### Competing Interests

The authors declare there are no competing interests.

### Author Contributions

- Yu Bao performed the experiments, analyzed the data, prepared figures and/or tables, authored or reviewed drafts of the paper, and approved the final draft.
- Bao Qi conceived and designed the experiments, performed the experiments, analyzed the data, prepared figures and/or tables, authored or reviewed drafts of the paper, and approved the final draft.
- Wei Huang performed the experiments, analyzed the data, authored or reviewed drafts of the paper, and approved the final draft.
- Bao Liu conceived and designed the experiments, analyzed the data, prepared figures and/or tables, authored or reviewed drafts of the paper, and approved the final draft.
- Yu Li conceived and designed the experiments, prepared figures and/or tables, authored or reviewed drafts of the paper, and approved the final draft.

### Data Availability

The original Illumina sequencing data generated in this study is available at NCBI SRA: PRJNA523683.

### Supplemental Information

Supplemental information for this article can be found online at http://dx.doi.org/10.7717/peerj.9930#supplemental-information.

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
