# Peer review of "The fungal community in non-rhizosphere soil of Panax ginseng are driven by different cultivation modes and increased cultivation periods"

_PeerJ, doi:10.7717/peerj.9930_

## Round 0.1 · original submission · Major Revisions

· Academic Editor

Major Revisions

Dear Dr Bao and co-authors,

I have received two reviews from experts in the discipline field. Whilst both reviewers can see the potential in your study, both state that major changes need to be made before this manuscript can be reconsidered in the review process.

As such, I would request that you take on board all of the comments and respond to each on individually and in detail so as to highlight the changes that you decide to make.

Firstly, many part of the manuscript text are not grammatically correct (e.g. Abstract, Introduction, Discussion and Conclusion) and this means that the whole manuscript requires thorough editing by somebody experienced in written English science communication, OR alternatively the PeerJ editing service.

Consider changes to the title to better reflect the treatments in your study (see R2).

A strong rationale or justification for why this study is needed and strong research questions (or hypotheses) that are being answered by the study are required. These will provide you with structure for your in reporting the results and then discussing the results, and finally stating in the conclusions whether the research questions were answered and what you found in response to each research question

You must provide greater detail on the methods used and justify the decisions made in site selection, treatments etc.

The Discussion should not simply reiterate the results - it should propose logical and supported theories as to why these results were observed and what they mean - often placing your findings int he context of similar or contrasting research.

Good luck in providing a thorough and major revision of this manuscript.

Regards

Steve

Reviewer 1 ·

Basic reporting

- Some parts (for example, L83-86, 213-215, 253-255, 258-259) are not grammatically correct or sounds awkward. Please revise.

Experimental design

- Please provide more information on how the ginseng fields were selected.

Validity of the findings

- L253-255 in the Conclusion is not clear.

Additional comments

1. Please make it clearer why fungal communities in non-rhizosphere soil were studied. What did the studies conducted on rhizosphere soils find? What is the importance of non-rhizosphere soils? Providing answers to these questions would help the readers better understand the rationale of this study.

2. The Discussion is mainly reiteration of results, and thus needs significant improvement. Literature on studies of soils from ginseng fields or other crops need to be incorporated. Comparison of previous studies with current study will enrich the Discussion.

3. Please use alternative terms for 'artificial cultivation' and 'soil micrrobiome community imbalance'. They are confusing terms.

4. The figure legends need to be more specific. Please indicate what C2, C5, W15, W35 are, even if they were explained in the Materials and Methods section. This will help the readers better understand the contents in the figures.

Reviewer 2 ·

Basic reporting

no comment

Experimental design

no comment

Validity of the findings

no comment

Additional comments

This ms studied the impact of cultivation models, period and fallowing on fungal community composition in the non-rhizosphere compartment of ginseng soil. The current ms did not use the data effectively. Current result cannot support the objective of this study perfectly. Introduction, discussion and conclusions also need be improved. There are a lot of minor mistakes in current ms, the authors should be more careful.
My specific comments are as follows:
Title: present title is not appropriate. There are five treatments in your ms, which also include different cultivations and cultivation period. Only considering fallowing is not appropriate.

Abstract:
Lines 22-28 part of background is too long, please shorten them. Meanwhile, only fungal community composition cannot represent or reflect soil microbiome imbalance. The objective of measuring fungal community composition is not clear, reconsider it please.

Lines 28-31 Introduce your treatments more clearly, this can help readers understand your research quickly. (cultivation models, cultivation period; W15, W35, C2, C5, F10)

Lines 38-40 conclusion is not well written. I think that you cannot say fallowing is a practical and effective practice to restore healthy fungal community only according to the results of OTU richness and PD whole tree.

Introduction:
Lines 47-50 As you cited here (It has been reported that replantation of ginseng 50 may cause up to 75% of ginseng seedling death), continuous cropping obstacles is a limitation of yield and quality of Ginseng. Thus, soil microbial community imbalance is a factor of continuous cropping obstacles. For soil microbial community imbalance, the growth of pathogenic fungi is a main factor of yield decrease of Ginseng. I suggest that add some data of Ginseng yield and quality, and then do some functional prediction analysis using FUNGuild (http://www.stbates.org/guilds/app.php). Combination of the results of alpha diversity and functional analysis can represent soil microbial community imbalance effectively. These will improve this ms a lot.

Lines 57-60 All right, but you did not offer the result of bacteria in this ms.

Lines 69-74 Objective is not well written, reconsider and improve it please.

Materials and methods:
Line 78 As far as I known, you only collected soil samples for one time. So, why you chose this season? And in which day in this month (July 2016)? Make this more clear, please.

Line 79 Add the longitude and latitude here.

Lines 86-88 This is repeating Line 82.

Line 91 How do you prevent cross contamination of different soil samples by using PVC tube?

Lines 92-93 How much soil do you use for DNA extraction? Meanwhile, do you have negative control? See Schöler et al. (2017) and Vestergaard et al. (2017) in Biology and fertility of soils, please.

Line 103 QIIME1 or QIIME2? Make this clear, please. QIIME 1 is no longer supported at this time, as development and support effort for QIIME is now focused entirely on QIIME 2 (http://qiime.org/).

Line 114 Add the method of PCoA analysis and T test.

Results:
Lines 124-125 Delete, this is repeating Lines 69-71.

Lines 126-130 Delete, this is repeating Lines 80-83.

Line 130 Whole name for ITS should be used in Line 95.

Line 135 Add the whole name of PCoA here. Meanwhile. I suggest doing Adonis or anoism analysis combined with PCoA, this could give more information of your data.

Line 166 R2, revise it please.

Discussion is not well written, after revising your ms (especially data analysis, functional analysis and result part), improve discussion as well.

Conclusions
Conclusions are not well written. Current result cannot reflect fungal community imbalance as you said in abstract and introduction.

References
Lines 291-292 “…Study on fungi diversity of ginseng rhizosphere soil in northeastern China…” only capitalize the first letter of “Study”.

Lines 302-303 same as Lines 291-292

Line 337 …soil fungal global…

Lines 339-341 same as Lines 291-292

Lines 355-359 same as Lines 291-292

Table and Figures
Figure 1A The PCoA analysis is done based on OTU data or genus data? Clarify this please. Is there 0 value in your data? Was the data Hellinger transformed when you do PCoA analysis? And do adonis and anoism analysis to combined with PCoA analysis, this can give more information to readers.

Figure 1B where is F10 treatment in this Venn diagram.

Figure 1C Phylum level can give us little information. Thus, I suggest to use Class level to make this figure. Meanwhile, do the statistic (ANOVA or Kruskal-Wallist, it is up to your data) analysis for each taxa.

Figure 2 R2, not R2. Keep preciseness.

Table 1 Add the indication of W15, W35, C2, C5 and F10 in the note of Table. Use mean value of three replications and add SE or SD value. Then, do statistic analysis for five treatments. Meanwhile, did you do the normality and homogeneity of variance analysis of your data?

---

## Round 0.2 · accepted · Accept

· Academic Editor

Accept

Good work on the revisions, Rev 1 was very content, as am I.
Congratulations.

Reviewer 2 ·

Basic reporting

After revised, the manuscript improved a lot, and is suitable for publishing in PeerJ.

Experimental design

well

Validity of the findings

no comments

Additional comments

no comments